

# An investigation of the pigments, antioxidants and free radical scavenging potential of twenty medicinal weeds found in the southern part of Bangladesh

Mousumi Jahan Sumi[1], Samia Binta Zaman[2], Shahin Imran[3,4], Prosenjit Sarker[1], Mohammad Saidur Rhaman[5], Ahmed Gaber[6], Milan Skalicky[7], Debojyoti Moulick[8] and Akbar Hossain[9]

[1] Department of Crop Botany, Khulna Agricultural University, Khulna, Bangladesh
[2] Faculty of Agriculture, Khulna Agricultural University, Khulna, Bangladesh
[3] Department of Agronomy, Khulna Agricultural University, Khulna, Bangladesh
[4] Institute of Plant Science and Resources, Okayama University, Kurashiki, Japan
[5] Department of Seed Science and Technology, Bangladesh Agricultural University, Mymensingh, Bangladesh
[6] Department of Biology, Faculty of Science, Taif University, Taif, Saudi Arabia
[7] Department of Botany and Plant Physiology, Faculty of Agrobiology, Food, and Natural Resources, Czech University of Life Sciences Prague, Prague, Czech Republic
[8] Department of Environmental Science, University of Kalyani, Nadia, West Bengal, India
[9] Division of Soil Science, Bangladesh Wheat and Maize Research Institute, Dinajpur, Rangpur, Bangladesh

Corresponding authors
Shahin Imran,
shahinimran124@gmail.com
Akbar Hossain,
akbarhossainwrc@gmail.com

## ABSTRACT

Despite their overlooked status, weeds are increasingly recognized for their therapeutic value, aligning with historical reliance on plants for medicine and nutrition. This study investigates the medicinal potential of native weed species in Bangladesh, specifically pigments, antioxidants, and free radical scavenging abilities. Twenty different medicinal weed species were collected from the vicinity of Khulna Agricultural University and processed in the Crop Botany Department Laboratory. Pigment levels were determined using spectrophotometer analysis, and phenolics, flavonoids, and DPPH were quantified accordingly. Chlorophyll levels in leaves ranged from $216.70 \pm 9.41$ to $371.14 \pm 28.67$ µg g$^{-1}$ FW, and in stems from $51.98 \pm 3.21$ to $315.89 \pm 17.19$ µg g$^{-1}$ FW. Flavonoid content also varied widely, from $1{,}624.62 \pm 102.03$ to $410.00 \pm 115.58$ mg CE 100 g$^{-1}$ FW in leaves, and from $653.08 \pm 32.42$ to $80.00 \pm 18.86$ mg CE 100 g$^{-1}$ FW in stems. In case of phenolics content *Euphorbia hirta* L. displaying the highest total phenolic content in leaves ($1{,}722.33 \pm 417.89$ mg GAE 100 g$^{-1}$ FW) and *Ruellia tuberosa* L. in stems ($977.70 \pm 145.58$ mg GAE 100 g$^{-1}$ FW). The lowest DPPH $2.505 \pm 1.028$ mg mL$^{-1}$ was found in *Heliotropium indicum* L. leaves. Hierarchical clustering links species with pigment, phenolic/flavonoid content, and antioxidant activity. PCA, involving 20 species and seven traits, explained 70.07% variability, with significant PC1 (14.82%) and PC2 (55.25%). Leaves were shown to be superior, and high-performing plants such as *E. hirta* and *H. indicum* stood out for their chemical composition and antioxidant activity. Thus, this research emphasizes the value of efficient selection while concentrating on the therapeutic potential of native weed species.

## INTRODUCTION

Plants have long been valued for their abundant primary and secondary metabolites, which have significant medicinal and nutraceutical benefits. Historically, numerous nations, ethnic groups, and societies have used these botanical resources (*Ghosh et al., 2019a*; *Ghosh et al., 2019b*). Natural remedies, particularly medicinal plants, were the primary means of treating illnesses, and their raw materials were critical to the pharmaceutical industry (*Weckesser et al., 2007*; *Zargoosh et al., 2019*). According to the World Health Organization, till now approximately 80% of the global population prefers herbal extracts in primary health care due to their potential to replace chemical drugs with fewer side effects (*Noguchi & Niki, 2000*; *Weckesser et al., 2007*; *Zargoosh et al., 2019*). Weeds are wild plants that grow spontaneously, often without human cultivation, and are commonly found around residential areas or farmlands (*Rizki, Nursyahra & Fernando, 2019*). They reproduce vigorously beyond their natural habitat, adversely affecting main crops by competing for light, nutrition, and space, resulting in reduced crop growth and productivity (*Riaz et al., 2007*; *Singla & Pradhan, 2019*). Hence, weed management becomes imperative. However, the overuse of synthetic pesticides and herbicides to remove weeds poses risks of toxicity to both humans and the environment. This emphasizes the need for environmentally friendly alternatives, such as using weeds for cost-effective solutions (*Sribanditmongkol et al., 2012*; *Wiwanitkit, 2013*; *Rattanata et al., 2014*).

Common weeds, known for their natural resistance to microbial attacks (*Rattanata et al., 2014*). There is huge evidence where weeds are being used for human health benefits. For instance, *Acalypha indica* L. (anti-tuberculosis, respiratory disorder and antibacterial activity) (*Govindarajan et al., 2008*; *Gupta et al., 2010*), *Alternanthera sessilis* L. (improve eye health, high in antioxidants, cooling effect on body and eyes, and neuroprotective) (*Walter, Merish & Tamizhamuthu, 2014*), *Bryophyllum calycinum* S. (antimicrobial, antihypertensive, activity immunomodulatory activity, anti-inflammatory, analgesic activity, gastrointestinal activity, anti-diarrheal, antihistaminic effect, anti-diabetic activity and cardiovascular activity) (*Pandurangan, Kaur & Sharma, 2015*), *Coccinia grandis* L. (analgesic, anti-inflammatory, and anticancer effects) (*Pekamwar, Kalyankar & Kokate, 2013*), *Eclipta postrata* L. (use in dermatological, hepatic, and gastric disorders) (*Timalsina & Devkota, 2021*), *Enhydra fluctuans* Lour. (analgesic, anti-inflammatory, antimicrobial, anticancer, neuroprotective) (*Barua et al., 2021*), *E. hirta* (respiratory disorders, COVID-19 medication, asthma) (*Cayona & Creencia, 2022*; *Gupta et al., 2017*), *H. indicum* (utilized for antiemetic, amenorrhea, failure-to-thrive in infants, ocular infections, and hypertension) (*Togola et al., 2005*), *Oxalis corniculata* L. (antibacterial effect) (*Mukherjee et al., 2013*), *Parthenium hysterophorus* L. (traditional remedy for dermatological conditions, ulcerated sores, facial neuralgia, fever, and anemia) (*Venkataiah et al., 2003*; *Patel, 2011*), *Persicaria lapathifolia* L. (antioxidant and allelopathic activities) (*Abd-El Gawad et al., 2021*) *etc.* Also, some weeds have antifungal agent *e.g.*, *Chenopodium album* L. has antifungal effect

on *Fusarium* sp. (*Waqas, Akbar & Andolfi, 2024*); and some weeds are used to isolate herbicidal compound (*Akbar et al., 2023*; *Akbar et al., 2022*).

Plant pigments, whether main or accessory, have great potential for use in preventing and treating diseases. They act as antioxidants and can help fight cancer (*Redzić, Hodzić & Tuka, 2005*). Moreover, antioxidants protect the body from the harmful effects of nitrogen, chlorine, and reactive oxygen species, thereby preventing disease (*Zaveri, 2006*; *Zargoosh et al., 2019*). Reactive oxygen species (ROS), including radicals such as hydroxyl, nitric oxide, hydrogen peroxide, and superoxide, are produced during normal metabolic activities. Abnormal functioning or low antioxidant levels lead to oxidative stress, which contributes to degenerative diseases such as Parkinson's disease, cancer, Alzheimer's disease, and diabetes (*Hazra, Biswas & Mandal, 2009*; *Jafri et al., 2022*). Antioxidants work by either accepting or donating electrons to neutralize free radicals, thereby reducing their ability to harm biomolecules. This process generates less active new free radicals, which are then neutralized to effectively stop the chain reaction (*Halliwell, 2007*; *Sharma & Kharel, 2019*). Medicinal weeds offer an alternative source of antioxidants, especially those rich in phenolic and flavonoid compounds. These antioxidants play a crucial role in protecting against oxidative damage and a variety of illnesses, such as neurological, immunological, and cardiovascular diseases (*Kumar & Kumar, 2009*; *Ceccanti et al., 2018*).

However, while many weeds have been known for their potential medicinal benefits, comprehensive studies exploring their pigments, antioxidants, and free radical scavenging potential remain limited, especially in the context of Bangladesh. Recognizing the significant role of antioxidants in preventing various diseases and the increasing global preference for natural remedies, there is a pressing need to explore the medicinal properties of common weeds indigenous to Bangladesh. Furthermore, with worries increasing about chemicals used in farming, like pesticides and herbicides, there is an urgent need for eco-friendly alternatives, like using medicinal weeds instead. Therefore, this study examines twenty common weeds in Bangladesh to understand their pigments, antioxidants, and potential to fight off harmful free radicals. Focusing on the abundant weeds in the southern region, it aims to uncover their medicinal properties and support the development of nutraceutical industries. In order to create new therapeutic approaches, it is essential to do more study in this field to clarify the precise mechanisms behind the therapeutic qualities of these plant species.

## MATERIALS & METHODS

### Sampling and data collection

The twenty different species (Table S1) of medicinal weed plants were chosen (consulting with Prof. Dr. Yeamin Kabir, Khulna University) for this investigation between June and July 2023. The most well-known native type weed species were chosen for our experiment. The selected weed plants, about three months old, were gathered from around the Khulna Agricultural University campus in Khulna and promptly put in zip-lock bags and stored in an ice box. After being gathered, the samples were taken to the Crop Botany Department Laboratory at Khulna Agricultural University for further study. Upon arrival at the

laboratory, the collected samples were meticulously processed by separating the leaves from the stems. The prominent midribs were carefully removed from the leaves, and then the plant parts were combined to create composite samples for further analysis. The leaves and stems of each of the twenty plant species under investigation were then chopped to prepare working samples for pigment analysis. In order to get data on antioxidants, another portion of fresh leaf and stem material was extracted using a morter and pastle. In a completely randomized design (CRD), there were 40 treatment combinations. Each treatment had four replications, with each replication consisting of four distinct plants. This was done for each species of plant being studied.

## Determination of pigments

The present protocol outlines the methodology for determining the levels of chlorophyll *a*, chlorophyll *b*, total chlorophyll, and total carotenoids in a whole pigment extract of green plant tissue using a spectrophotometer. The protocol was slightly modified after Lichtenthaler (*Lichtenthaler, 1987*). A total of 200 µL of distilled water were added to 50 mg of freshly chopped composite leaf and stem samples, which were subsequently placed in glass bottles. After through shaking, 16 mL of ethanol was added, and the mixture was then left in an airtight condition in the dark for 24 h. The following day, the absorbance was read at 470, 649, 666 and 750 nm wavelengths using a spectrophotometer (Shimadzu UV-1280; Shimadzu, Kyoto, Japan). The following formulas were subsequently used to determine the total amount of chlorophyll (*a*, *b* and *a* +*b*) and carotenoids:

$$\text{Chlorophyll } a(C_a) = \frac{(13.36 \text{ A666} - 5.19 \text{A649}) \times 16.2}{FW}$$

$$\text{Chlorophyll } b(C_b) = \frac{(27.43 \text{ A649} - 8.12 \text{ A666}) \times 16.2}{FW}$$

$$\text{Total chlorophyll} = \text{Chlorophyll } a(\text{Ca}) + \text{Chlorophyll } b(\text{Cb})$$

$$\text{Carotenoids} = \frac{(4.785 \text{ A470} + 3.657 \text{ A666} - 12.76 \text{ A649}) \times 16.2}{FW}$$

where, A649 = absorbance at 649 nm, A664 = absorbance at 664 nm, A470 = absorbance at 470 nm, and FW = fresh weight of plant tissue extracted (mg).

## Total phenolic content assay

To quantify the total phenolic compound content, a technique adapted from the research of *Albano & Miguel (2011)*. The 3 g fresh leaf and stem samples were homogenized in 30 mL of 99.9% ice-cooled methanol using a mortar and pestle. The mixture was then placed in an airtight glass bottle and kept in the dark for 30 min. The supernatant extracts were collected into two different 1.5 mL eppendorf tubes. The tubes were then centrifuged at 15,000 rpm for 5 min and stored in a refrigerator at −10 °C for the quantification of phenol, flavonoid, and DPPH radical scavenging capacity. For total phenolics determination gallic acid was used as a standard. A total of 330 µL various concentrations of gallic acid or plant extracts were added to a 50 mL test tube. Subsequently, 16 µL Folin–Ciocalteu reagent, and 3 mL of 10% $Na_2CO_3$ solution was added, and the mixture was left in the dark at room temperature for 30 min. At a wavelength of 760 nm,

the absorbance of each sample was measured, indicating the overall phenolic content of the compounds. These absorbance values were plotted on the $y$-axis against their corresponding concentrations on the $x$-axis. This plot generated a linear relationship, forming a standard curve that was used to determine the total phenolic content in the test samples.

## Determination of total flavonoid content

The flavonoid contents of the leaf and stem extracts were calculated using the catechinas standard. A total of 300 µL AlCl$_3$ and 300 µL NaNO$_2$ solutions were reacted with 1 mL previously prepared plant extracts. The mixture was incubated for 5 min at room temperature. After that, 2 mL NaOH solution and 10 mL distilled water was added. The prepared solution was vortexed, after which the mixture was allowed to stand for 30 min. Finally, the absorbance was measured at 510 nm. The absorbance value represents the total flavonoid content (TFC) of the compound, as described by *Baba & Malik (2015)*. The TFC was measured as µg of equivalent catechin per gram of fresh extract.

## DPPH radical scavenging capacity assay

The 2-dipheny l-l-picrylhydrazyl radical (DPPH) was used to test the medicinal plant extracts' capacity to scavenge free radicals (*Brand-Williams, Cuvelier & Berset, 1995*). DPPH is a stable free radical with a characteristic of violet color. The sample turns yellow as a result of the antioxidants scavenging the free radicals in it. The amount of radical scavenging activity is directly correlated with the color shift from violet to yellow. Various concentrations of plant extracts (prepared using previously extracted material and methanol) were combined with 1 mM DPPH in 300 µL methanol. After vortexing to thoroughly mix the ingredients, the samples were allowed to incubate at room temperature (24–30 °C) for 30 min. A DR 6000 UV spectrophotometer was used to measure the extent to which the absorbance decreased at 517 nm. The percentage of scavenging activity was determined as follows:

$$\text{IC\_50} = \frac{(\text{Ac} - \text{As}) \times 100}{\text{Ac}}$$

where 'As' is the sample absorbance and 'Ac' being the control absorbance (without extract). To determine the IC$_{50}$ value, the percentage of radical scavenging activity was plotted against the extract's corresponding concentration. The amount of antioxidant material needed to scavenge 50% of the free radicals in the assay system is known as the IC$_{50}$. According to *Nisha, Nazar & Jayamurthy (2009)* the IC$_{50}$ values and antioxidant activity are inversely correlated.

## Statistical analysis

The statistical analysis was performed using Mini Tab 17.3. One-way ANOVA was used to identify significant differences, and Tukey's HSD test ($p \leq 0.05$) was subsequently performed. The "pheatmap" package in R 4.3.2 (*R Core Team, 2024*) was used to create a heatmap and carry out hierarchical clustering analysis using Euclidean distances. The "GGally" and "factoextra" packages were used to perform principal component analysis (PCA) (*Rabbi et al., 2024*; *Imran et al., 2023*). The best performer was selected using the "RhCoClust" algorithm package (*Mohi-Ud-Din et al., 2021*).

## RESULTS

### Mean variability in plant traits

In the experimental setup involving 20 medicinal weed species and their respective plant parts, significant variations were observed both within plant parts and among species for all parameters studied. The descriptive statistics of the species traits are presented in the box plot (Fig. 1). For chlorophyll *a*, the leaf exhibited a minimum concentration of 216.70 µg g$^{-1}$ FW, Quartile 1 had 242.16 µg g$^{-1}$ FW, a median of 255.51 µg g$^{-1}$ FW, Quartile 3 had 300.21 µg g$^{-1}$ FW, and a maximum of 371.14 µg g$^{-1}$ FW, whereas the stem concentration ranged from 51.98 to 315.89 µg g$^{-1}$ FW (Fig. 1A). In the case of chlorophyll *b*, the leaf values ranged from 106.82 to 301.19 µg g$^{-1}$ FW, with corresponding stem values ranging from 33.69 µg g$^{-1}$ FW to 180.66 µg g$^{-1}$ FW (Fig. 1B). The total chlorophyll concentration exhibited a broad range, with leaf concentrations ranging from 368.34 to 621.91 µg g$^{-1}$ FW and stem concentrations ranging from 86.08 to 496.55 µg g$^{-1}$ FW (Fig. 1C). The carotenoid concentrations in the leaves ranged from 3.65 to 104.88 µg g$^{-1}$ FW, and those in the stems ranged from 11.89 to 80.46 µg g$^{-1}$ FW (Fig. 1D). Phenolic and flavonoid values also demonstrated significant differences between leaf and stem tissues (Figs. 1E & 1F). With one notable exception in terms of the IC$_{50}$, all the stem parameters exhibited significant decreases. In comparison to those of stems, the concentrations of carotenoids, total phenolics, flavonoids, chlorophyll *a*, and chlorophyll *b* were 2.33, 2.56, 2.44, 2.47, 3.57, and 10.18 times *greater*, respectively (Fig. 1). On the other hand, the IC$_{50}$ value of the leaves was 1.62 times lower than that of the stems, indicating that the leaves had greater potential for antioxidant scavenging than the stems did (Fig. 1G).

### Chlorophyll *a* content in the leaves and stems of twenty medicinal weed species

The chlorophyll *a* content in the leaves and stems of the twenty different medicinal weed species tested varied significantly. *H. indicum* and *S. dulcis* have the highest chlorophyll *a* percentage in both their leaves and stems, at 100%. *A. indica* has relatively high chlorophyll *a* level in both leaves (90.56%) and stems (56.97%). *A. sessilis* and *Bryophytllum calycinum* S. also show significant leaf chlorophyll *a* concentration, with 80.79% and 65.15%, respectively, but lower stem values (Fig. 2A). Most species exhibit higher chlorophyll *a* concentration in leaves than in stems, with notable exceptions like *Ageratum conyzoides* L., which shows a higher stem chlorophyll *a* value (81.82%) compared to its leaf (67.18%). When solely analyzing plant species, *S. dulcis* yielded the highest yield at 305 µg g$^{-1}$ FW, while *B. calycinum* had the lowest yield at 146.889 µg g$^{-1}$ FW, regardless of the plant part (Table 1). When only plant parts alone were considered, the chlorophyll content was approximately 2.07 times greater in the leaves than in the stems, indicating that the leaves are more efficient than the stems.

### Chlorophyll *b* content in the leaves and stems of twenty medicinal weed species

The chlorophyll *b* content in the leaves and stems of the twenty different medicinal weed species differed widely. *A. indica* displays high chlorophyll *b* concentrations in both leaves

**Table 1  Effects of plant species and plant parts on several pigments, phytochemical properties and free radical scavenging potential in 20 medicinal weeds.** The data are presented as the means of 3 replicates ±SEs, with a sample size of $n = 3$. Statistical analysis was performed using one-way ANOVA with Tukey's *post hoc* test; different letters (a, b, c,...., *etc*) indicate statistically significant differences ($p < 0.05$).

| Plant species and parts | Chlorophyll *a* ($\mu g\,g^{-1}$ FW) | Chlorophyll *b* ($\mu g\,g^{-1}$ FW) | Total Chlorophyll ($\mu g\,g^{-1}$ FW) | Carotenoids ($\mu g\,g^{-1}$ FW) | Phenolic (mg GAE 100 $g^{-1}$ FW) | Flavonoid (mg CE 100 $g^{-1}$ FW) | $IC_{50}$ to scavenge DPPH (mg mL$^{-1}$) |
|---|---|---|---|---|---|---|---|
| *Acalypha indica* | 258.02[a–c] | 164.75[a–d] | 422.77[bc] | 75.86[a] | 217.57[e–g] | 541.92[d–f] | 31.49[c–f] |
| *Ageratum conyzoides* | 253.89[bc] | 150.58[a–f] | 404.47[b–d] | 69.005[a–c] | 120.66[fg] | 635.38[cd] | 54.51[b–d] |
| *Alternanthera sessilis* | 222.77[b–e] | 106.48[d–f] | 329.25[d–g] | 68.41[a–c] | 169.55[fg] | 245.00[g] | 76.36[b] |
| *Bryophyllum calycinum* | 146.89[g] | 143.19[a–f] | 290.08[e–g] | 19.11[fg] | 271.73[e–g] | 521.54[d–f] | 14.32[ef] |
| *Centella verticillata* | 182.56[e–g] | 156.42[a–e] | 338.98[d–g] | 28.87[d–g] | 403.52[d–f] | 305.38[fg] | 26.22[d–f] |
| *Coccinia grandis* | 231.99[b–d] | 133.80[c–f] | 365.79[c–e] | 69.93[a–c] | 208.12[e–g] | 468.85[d–g] | 26.75[d–f] |
| *Eclipta postrata* | 161.18[g] | 165.96[a–d] | 327.14[d–g] | 19.19[fg] | 355.72[d–g] | 543.46[d–f] | 37.64[c–e] |
| *Enhydra fluctuans* | 151.43[g] | 126.61[c–f] | 278.04[fg] | 27.73[e–g] | 358.47[d–g] | 840.00[bc] | 31.63[c–f] |
| *Euphorbia hirta* | 212.85[c–f] | 143.96[a–f] | 356.81[c–f] | 56.36[a–f] | 1,089.93[a] | 468.85[d–g] | 15.08[ef] |
| *Heliotropium indicum* | 261.32[ab] | 141.44[a–f] | 402.76[b–d] | 66.63[a–d] | 511.49[c–e] | 390.38[d–g] | 7.26[f] |
| *Oxalis corniculata* | 256.92[bc] | 206.78[a] | 463.69[ab] | 52.95[a–f] | 166.15[fg] | 535.38[d–f] | 33.95[c–f] |
| *Parthenium hysterophorus* | 149.94[g] | 173.11[a–c] | 323.05[d–g] | 8.704[g] | 480.34[c–e] | 501.15[d–f] | 16.92[ef] |
| *Persicaria lapathifolia* | 168.19[f–g] | 89.32[f] | 257.51[g] | 49.38[a–f] | 729.63[bc] | 1,006.54[ab] | 6.86[f] |
| *Portulaca oleracea* | 163.23[g] | 142.79[a–f] | 306.02[e–g] | 24.003[fg] | 92.13[g] | 305.38[fg] | 58.26[bc] |
| *Ruellia tuberosa* | 168.32[f–g] | 108.69[c–f] | 277.01[eg] | 47.11[a–f] | 833.77[ab] | 990.00[ab] | 14.77[ef] |
| *Scoparia dulcis* | 305.0[a] | 205.63[ab] | 510.63[a] | 62.75[a–e] | 345.74[d–g] | 633.85[cd] | 29.55[c–f] |
| *Senna occidentalis* | 183.82[e–g] | 95.99[ef] | 278.8[fg] | 72.24[ab] | 363.95[d–g] | 1,120.0[a] | 36.87[c–f] |
| *Synedrella nodiflora* | 175.77[e–g] | 140.68[b–f] | 316.45[e–g] | 35.81[b–g] | 94.84[g] | 558.08[cd] | 71.66[b] |
| *Trema orientalis* | 188.86[d–g] | 115.92[c–f] | 304.78[e–g] | 47.11[a–f] | 149.23[fg] | 510.38[d–f] | 122.36[a] |
| *Tridax procumbens* | 155.15[g] | 117.44[c–f] | 273.73[fg] | 33.38[c–g] | 644.07[b–d] | 975.38[ab] | 7.59[ef] |
| **Plant parts:** | | | | | | | |
| Leaf | 269.57[A] | 197.46[A] | 467.04[A] | 62.37[A] | 543.86[A] | 906.89[A] | 29.37[B] |
| Stem | 130.24[B] | 85.39[B] | 215.74[B] | 31.11[B] | 216.80[B] | 308.92[B] | 42.64[B] |

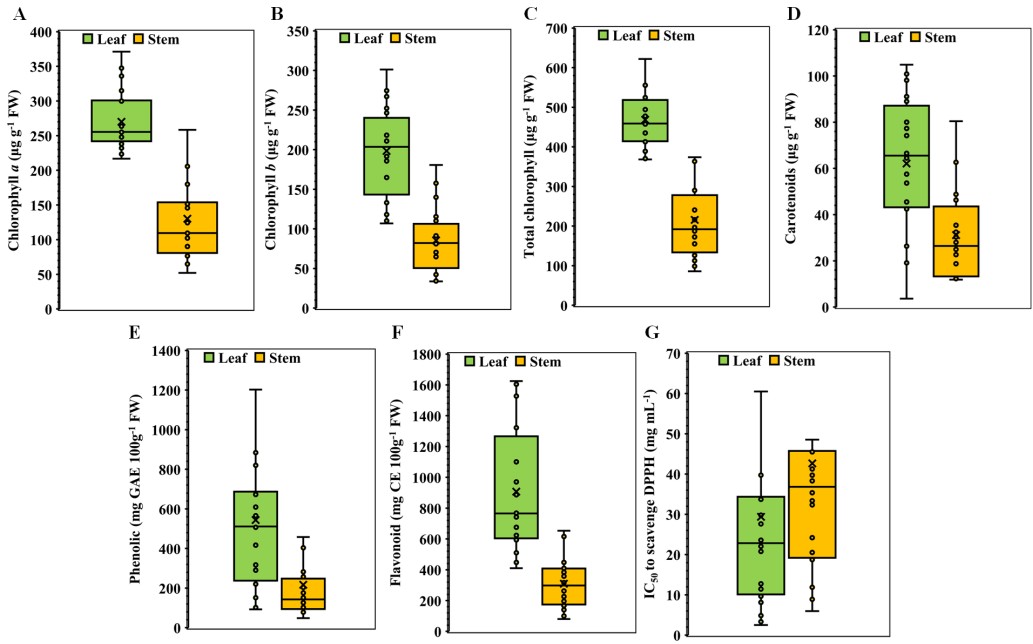

**Figure 1 Boxplots illustrating of descriptive statistics.** (A) Chlorophyll *a*, (B) chlorophyll *b*, (C) total chlorophyll, (D) carotenoids, (E) total phenolics, (F) total flavonoids, and (G) $IC_{50}$ value for scavenging DPPH free radicals in the leaves and stems of twenty selected medicinal weed species. The cross points are treatment means, and the horizontal lines dividing the box represent the medians. The higher and lower whiskers and the box boundaries at the top and bottom denote the Q3 (75th percentile), Q1 (25th percentile), maximum (Q1+1.5 IQR), and minimum (Q1 −1.5 IQR) values, respectively.

(72.83%) and stems (60.96%). *B. calycinum* has a high leaf chlorophyll *b* concentration (83.76%) but a significantly lower stem concentration (18.87%). *C. grandis* stands out with a lower leaf chlorophyll *b* percentage (36.54%) but a very high stem percentage (87.22%). *P. hysterophorus* shows the highest leaf chlorophyll *b* value (100%) while *S. dulcis* exhibits the highest stem chlorophyll *b* concentration (100%) (Fig. 2B). In general, most species have higher chlorophyll *b* concentrations in their leaves than in their stems, similar to the chlorophyll *a* data. However, exceptions exist, such as *O. corniculata* and *C. grandis*, where stem chlorophyll *b* concentrations are higher than those in leaves. According to the analysis of the individual plant species, *O. corniculata* yielded the highest yield at 206.779 $\mu g\,g^{-1}$ FW, while *P. lapathifolia* had the lowest yield at 89.315 $\mu g\,g^{-1}$ FW, irrespective of the plant part (Table 1). When focusing solely on plant parts, the chlorophyll content was approximately 2.31 times greater in leaves than in stems, highlighting the greater efficiency of leaves in chlorophyll production.

## Total chlorophyll content in the leaves and stems of twenty medicinal weed species

The total chlorophyll levels in both the leaves and stems exhibited considerable variation across the twenty medicinal weed species that were tested. The total chlorophyll content varied among the plant species, ranging from 621.91 ± 17.98 to 368.34 ± 0.035 $\mu g\,g^{-1}$ FW in the leaves of *O. corniculata* and *C. grandis*, respectively. *A. indica* (89.31%), *H. indicum*

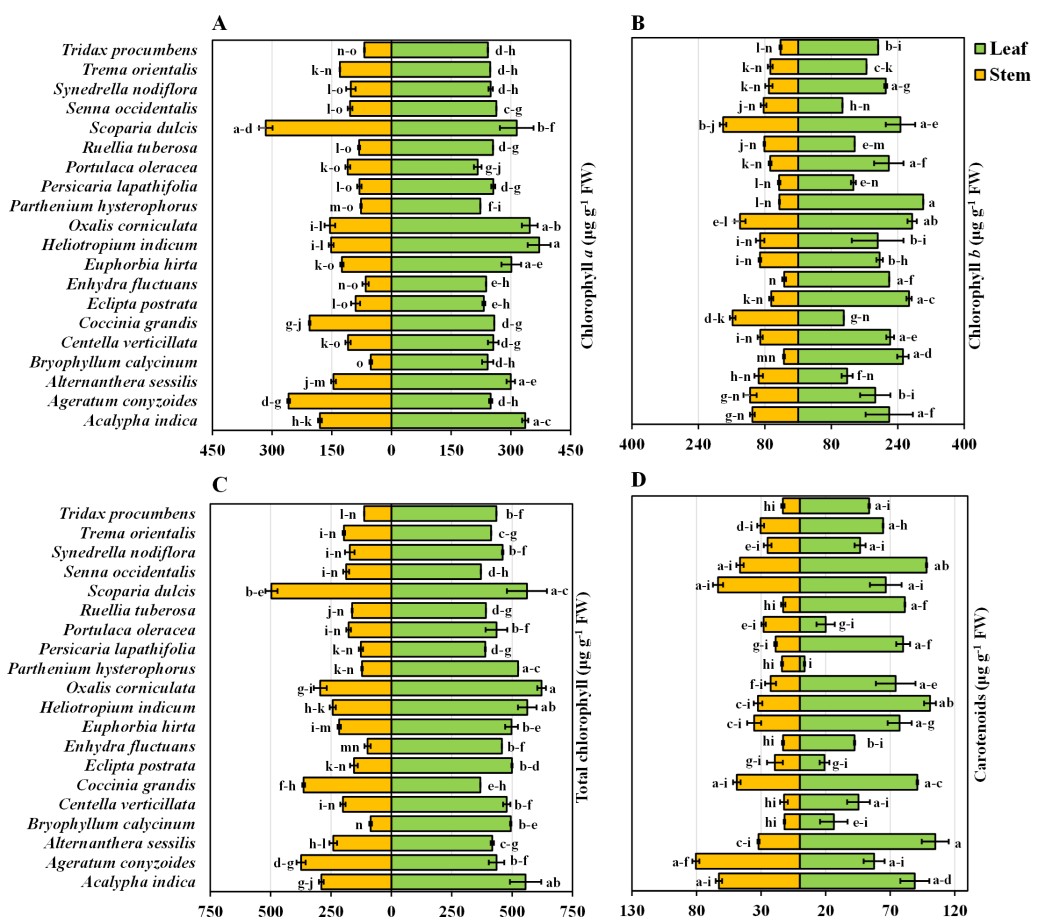

**Figure 2** **Bidirectional bar graph.** (A) Chlorophyll *a* content ($\mu$g g$^{-1}$ FW), (B) chlorophyll *b* content ($\mu$g g$^{-1}$ FW), (C) total chlorophyll content ($\mu$g g$^{-1}$ FW) and (D) carotenoid content ($\mu$g g$^{-1}$ FW) in the leaves and stems of twenty medicinal plants. The data are presented as the means of 3 replicates $\pm$SEs, with a sample size of $n = 3$. Statistical analysis was performed using one-way ANOVA with Tukey's *post hoc* test; different letters (a, b, c, ........., *etc.*) indicate statistically significant differences ($p < 0.05$).

(90.47%), and *O. corniculata* (100%) show exceptionally high chlorophyll content in leaves. *S. dulcis* (100%) and *C. grandis* (73.15%) also stand out with high chlorophyll levels, especially in stems. On the other hand, *B. calycinum* (17.34%) and *E. fluctuans* (19.86%) display relatively lower total chlorophyll concentrations in stems. Overall, *O. corniculata* leaves had the highest total chlorophyll content, and *B. calycinum* stems had the lowest total chlorophyll content when considering both leaves and stems across all the species. Among the individual plant species, *S. dulcis* had the highest content (510.63 $\mu$g g$^{-1}$ FW), while *P. lapathifolia* had the lowest (257.51 $\mu$g g$^{-1}$ FW), irrespective of the plant part (Table 1). When analyzing plant parts alone, the chlorophyll content was approximately 2.16 times greater in the leaves than in the stems.

## Carotenoids in the leaves and stems of twenty medicinal weeds

The carotenoid content in the leaves of *A. sessilis* and *P. hysterophorus* varied from 100% to 3.48%, respectively. In contrast, for the stems, the highest concentration of *A. conyzoides* was 100%, while the lowest concentration of *B. calycinum* was 14.77% (Fig. 2D). Additionally, the species with high total chlorophyll content include *A. indica* with 89.31% in leaves, *H. indicum* with 90.47% in leaves, and *S. dulcis* with 90.24% in leaves. In stems, *C. grandis* has a high total chlorophyll content of 73.15%. Overall, when considering both leaves and stems across all species, *A. sessilis* leaves exhibited the highest carotenoid values, and *P. hysterophorus* leaves showed the lowest carotenoid values. Among the individual plant species, *A. indica* had the highest yield at 75.86 $\mu$g g$^{-1}$ FW, while *P. hysterophorus* had the lowest yield at 8.704 $\mu$g g$^{-1}$ FW, irrespective of the plant part (Table 1). When focusing solely on plant parts, the carotenoid content in leaves was approximately 2.004 times greater than that in stems.

## Phenolic content in leaves and stems

The present study provided valuable information regarding the phenolic content of twenty important medicinal weeds. *E. hirta* stands out with the highest phenolic content in both leaves (100%) and stems (46.8%), suggesting a particularly rich source of these compounds. Other species like *P. lapathifolia* and *H. indicum* also show high phenolic content in their leaves (69.8% and 47.6%, respectively) and moderate to high levels in their stems. *R. tuberosa* is notable for its exceptionally high stem phenolic content (100%). The total phenolic content was highest in *E. hirta* leaves, while the lowest total phenolic content was obtained in the stems of *A. conyzoides*, *Synedrella nodiflora* L., *Trema orientalis* L., and *A. sessilis* (Fig. 3A). When two parts of the plant were taken into account, the phenolic content of the leaves was approximately 2.5 times greater than that of the stems. However, when only plant species were taken into account, *E. hirta* (1,089.93 mg GAE 100 g$^{-1}$ FW) reached its highest point, confirming its position as the dominant species, while *P. oleracea* (92.13 mg GAE 100 g$^{-1}$ FW) had the lowest value (Table 1).

## Flavonoid content in leaves and stems

Statistically, there was no significant difference among the leaves of *P. lapathifolia*, *S. occidentalis*, or *T. procumbens*. However, *S. occidentalis* exhibited the highest flavonoid content (100%), while *A. sessilis* had the lowest flavonoid content (25.24%) in the leaves (Fig. 3B). Among the stems, *A. sessilis* had the lowest flavonoid content (12.25%), while *R. tuberosa* had the highest (100%). Other species with high flavonoid content in leaves include *P. lapathifolia* (98.72%) and *T. procumbens* (94.03%). In stems, high flavonoid content was also observed in *S. occidentalis* (94.23%) and *T. procumbens* (64.78%) (Fig. 3B). Moreover, there were no significant differences in flavonoid content among the stems of *P. oleracea*, *E. postrata*, or *A. sessilis* (Fig. 3B). Considering the plant parts and species values, *S. occidentalis* leaves had the highest flavonoid content, while *A. sessilis* stems had the lowest. Regardless of plant part, when considering only plant species, *S. occidentalis* ranked highest (1,120.0 mg CE 100 g$^{-1}$ FW), and *A. sessilis* ranked lowest (245.00 mg CE 100 g$^{-1}$ FW) (Table 1). Moreover, when comparing plant parts, the leaves exhibited approximately 2.93 times greater flavonoid content than did the stems.

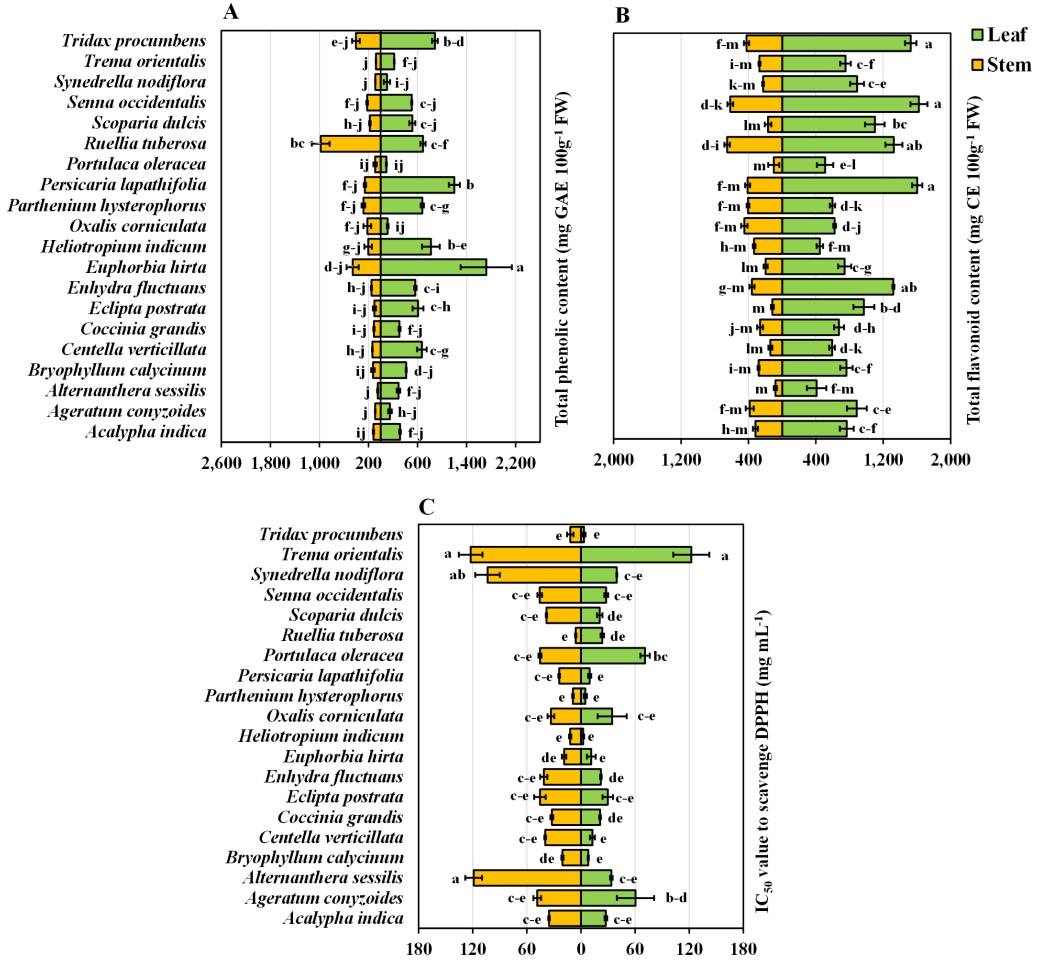

**Figure 3 Bidirectional bar graph.** (A) Total phenolic content (mg GAE 100 g$^{-1}$ FW), (B) total flavonoid content (mg CE 100 g$^{-1}$ FW), and (C) IC$_{50}$ value for scavenging DPPH free radicals (mg mL$^{-1}$) in the leaves and stems of twenty medicinal weeds. The data are presented as the means of 3 replicates ±SEs, with a sample size of $n = 3$. Statistical analysis was performed using one-way ANOVA with Tukey's post hoc test; different letters (a, b, c, ………., etc.) indicate statistically significant differences ($p < 0.05$).

## Antioxidative power in terms of DPPH radical scavenging capacity

In terms of DPPH radical scavenging capacity, *A. sessilis* leaves and *T. orientalis* stems have the lowest IC$_{50}$ values, set at 100%, indicating they have the highest antioxidant capacity, as they require the least amount of extract for effective radical scavenging. Conversely, *P. oleracea* exhibits the highest IC$_{50}$ values, at 4,886.05% for leaves and 2,053.22% for stems, making it the least effective antioxidant in this group. Other plants with strong antioxidant properties include *A. conyzoides*, with 132.02% for leaves and 199.30% for stems, and *S. dulcis*, with 192.75% for leaves and 149.27% for stems. When considering all the plant parts and species, no significant differences were observed in the leaves or stems of *T. orientalis* or the leaves of *A. sessilis*, both of which exhibited the highest values (Fig. 3C). However, when focusing solely on plant species, *T. orientalis* had the highest IC$_{50}$ value (122.36 mg mL$^{-1}$), while *P. hysterophorus* had the lowest (6.86 mg mL$^{-1}$) (Table 1). Furthermore,

when evaluating only plant parts, stems were found to possess $IC_{50}$ values 1.45 times greater than those of leaves. It is essential to note that a lower $IC_{50}$ value indicates greater antioxidant potential. Consequently, the antioxidant activity and radical scavenging ability of the leaves were generally superior to those of the stems.

## Hierarchical clustering and co-clustering of leaf genotypes

Hierarchical clustering heatmap illustrating the relationships among the leaves of twenty medicinal weed species based on various pigments, such as chlorophyll *a*, chlorophyll *b*, total chlorophyll, and carotenoids. Additionally, the analysis included free radical scavenging potential and important antioxidant properties, such as total phenolic and flavonoid contents, as shown in Fig. 4A. Row cluster 1 contained seven genotypes that were determined to be the most closely related, demonstrating an exceptionally high degree of similarity between these particular species. Closely behind, cluster 3 showed a significant assembly of six genotypes. However, clusters 4 and 5 each had all three of the remaining genotypes, whereas cluster 2 had only one (Fig. 4A). Again, within the three column clusters, each cluster contained a distinct combination of traits. Cluster 1, cluster 2 and cluster 4 each included two traits, while cluster 3 included only one trait. Notably, DPPH and carotenoids formed cluster 1, chlorophyll *a* and chlorophyll *b* clustered together in cluster 2, and cluster 4 comprised total chlorophyll and total phenolics. Additionally, the flavonoid content was distinctly distributed in cluster 3. The pigment values indicate that there are no noticeable variations among any of the clusters, as shown in the clustering bar graph (Fig. 4B). In Fig. 4B, the clustering bar graph provides a clearer depiction of the relationship between phytochemical contents and the species' ability to scavenge free radicals. Cluster 1 was superior to the other clusters in terms of phenolic content, with greater flavonoid content (Fig. 4B). Notably, both cluster 1 and cluster 2 exhibited lower $IC_{50}$ values than did cluster 3 and cluster 4, indicating a more potent ability to scavenge free radicals (Fig. 4B). Clusters 3 and 4, marked by lower flavonoid and total phenolic contents, exhibited reduced antioxidant capacity, evident in their higher $IC_{50}$ values. Conversely, cluster 5, displaying the lowest $IC_{50}$ value, demonstrated a significant rise in flavonoid content. Notably, antioxidants in both cluster 1 and cluster 2 showcased efficient free radical scavenging abilities.

## Hierarchical clustering and co-clustering of stem genotypes

The hierarchical clustering heatmap analysis focused on the stems revealed eight closely related genotypes in row cluster 3, indicating remarkable similarity among these specific species. In particular, major units of five and four genotypes were shown in clusters 1 and 5, respectively. In contrast, the remaining genotypes were dispersed across cluster 2, cluster 4, and cluster 6, each represented by a lone genotype (Fig. 5A). Within the four column clusters, distinct trait combinations characterized each cluster. Cluster-1 contained only one trait, while clusters 2, 3, and 4 each contained two traits.The graphical representation using bar graphs, with row clusters on the $X$-axis and trait values on the $Y$-axis, made it easy to grasp these trait combinations. Concerning pigments, clusters 1 and 2 were significantly different for all four pigments (Fig. 5B). This finding highlights the six species distributed in

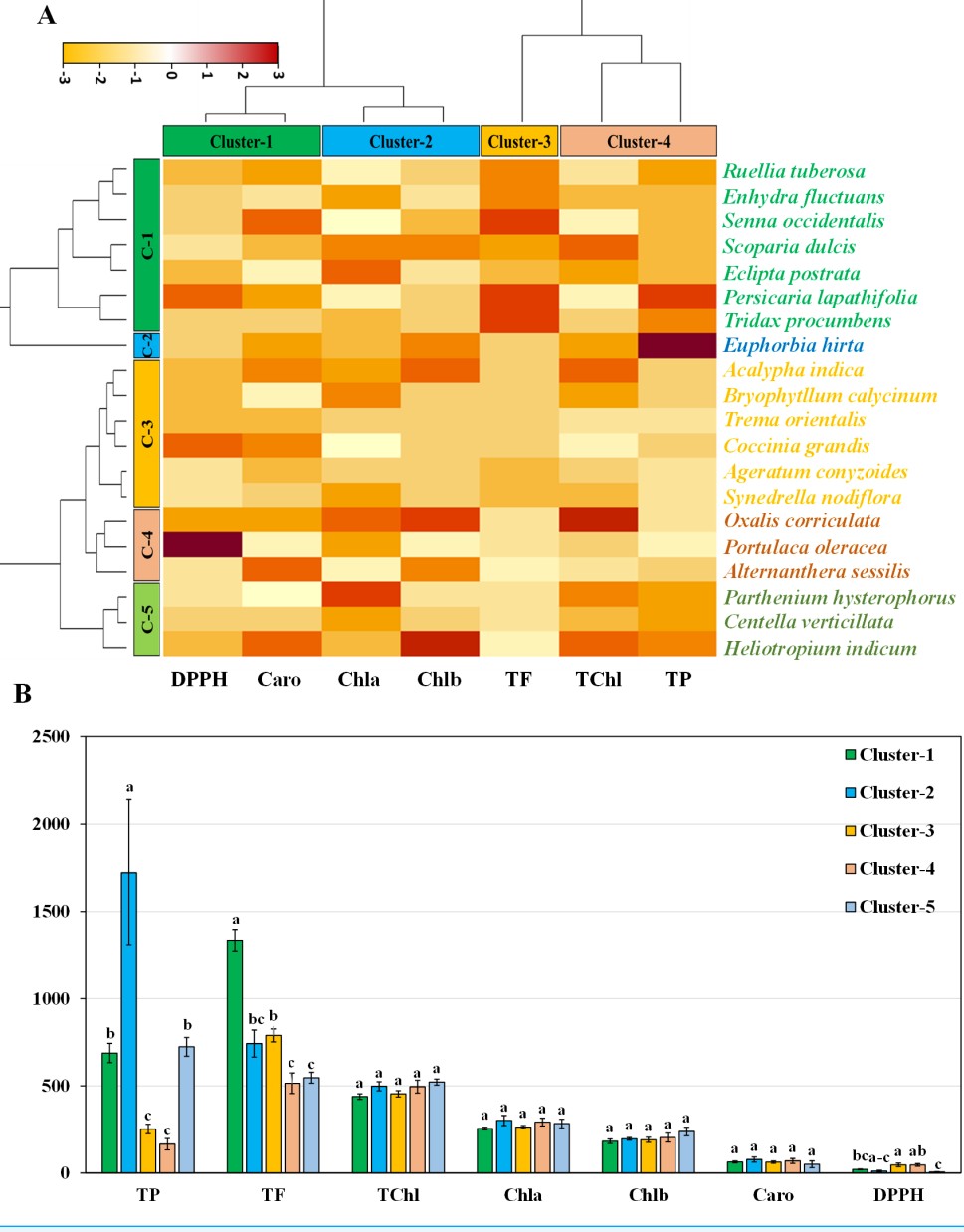

**Figure 4** **Hierarchical clustering and bar charts were generated to display the relationships between species and traits.** (A) Heatmap presenting scaled average values for seven leaf traits, with colours indicating the relative scale ($-3$ to $+3$) from the data standardization of pigments, antioxidants, and $IC_{50}$ values. The species and traits were grouped into six (rows) and four (columns) clusters, respectively. (B) Bar graph showing comparative cluster analysis of the leaf of selected species. The studied traits were Chla (chlorophyll $a$), Chlb (chlorophyll $b$), TChl (total chlorophyll), Caro (carotenoids), PC (phenolic), FC (flavonoid), and DPPH ($IC_{50}$ value for scavenging DPPH free radical). The data are presented as the means of 3 replicates $\pm$SEs, with a sample size of $n = 3$. Statistical analysis was performed using one-way ANOVA with Tukey's post hoc test; different letters (a, b, c, ………., *etc.*) indicate statistically significant differences ($p < 0.05$).

**Table 2  Best performers of 20 genotypes within different co-cluster combinations.**

| Co-Cluster Combinations | | Best Performers |
|---|---|---|
| RC-1 | CC-1 | *Heliotropium indicum* |
| | CC-2 | *Bryophyllum calycinum* and *Parthenium hysterophorus* |
| RC-2 | CC-1 | *Persicaria lapathifolia* |
| | CC-2 | *Tridax procumbens* and *Ruellia tuberosa* |
| RC-3 | CC-1 | *Euphorbia hirta* |
| | CC-2 | *Euphorbia hirta* |

Notes.
RC, row cluster; CC, column cluster; CC-1, Caro (carotenoids); Chla, (chlorophyll a); Chlb, (chlorophyll b); DPPH, (IC50 value for scavenging DPPH free radical); CC-2, FL (flavonoids); TChl, (total chlorophyll); TP, (phenolics).

cluster 1 and cluster 2 as being particularly noteworthy in terms of essential pigments. The clustering bar graph depicts the antioxidant activity and free radical scavenging potential measured in terms of the $IC_{50}$. Notably, cluster 4 contained the highest phenolic content, while clusters 5 and 6 exhibited the highest flavonoid content (Fig. 5B). The $IC_{50}$ values further revealed that the species in clusters 4, 5, and 6 exhibited lower $IC_{50}$ values, which is indicative of their increased antioxidant potential (Fig. 5B).

## Co-Cluster based selection of genotypes irrespective of plant parts

The robust hierarchical co-clustering method organizes rows, columns, and their relationships in datasets, even with outliers. In an effort to improve the accuracy and efficacy of the medicinal plant selection procedure, a co-cluster matrix was created using the robust co-cluster combinations under the RhCoClust algorithm. The clusters are organized into row clusters (RC-1, RC-2, RC-3), each further divided into two column clusters (CC-1 and CC-2) (Table 2). For instance, under RC-1, *H. indicum* excels in CC-1, while *B. calycinum* and *P. hysterophorus* stand out in CC-2. Similarly, in RC-2, *P. lapathifolia* leads in CC-1, with *T. procumbens* and *R. tuberosa* shining in CC-2. Finally, in RC-3, *E. hirta* emerges as the top performer in both CC-1 and CC-2.

## Correlation analysis

Pearson correlation analysis was used to investigate the relationships among twenty medicinal weed species traits, revealing significant interconnections among them (Fig. 6). The analysis covered various attributes, including chlorophyll *a* (Chla), chlorophyll *b* (Chlb), total chlorophyll (TChl), carotenoids (Caro), total phenolic content (TP), total flavonoid content (TF), and the $IC_{50}$ value for scavenging free radicals (DPPH) in both leaves and stems. Significant correlations were found among most pigments and phytochemical traits, except for Caro and Chlb, TP and Chlb, and TP and Caro. While there were nonsignificant negative correlations with the other parameters, the radical scavenging potential (DPPH) showed significant negative correlations with the Chla concentration. These findings, imply that the contents of both investigated phytochemicals increase as the pigment content increases, and vice versa. However, as the DPPH concentration rises, the concentrations of each pigment and antioxidant decrease.

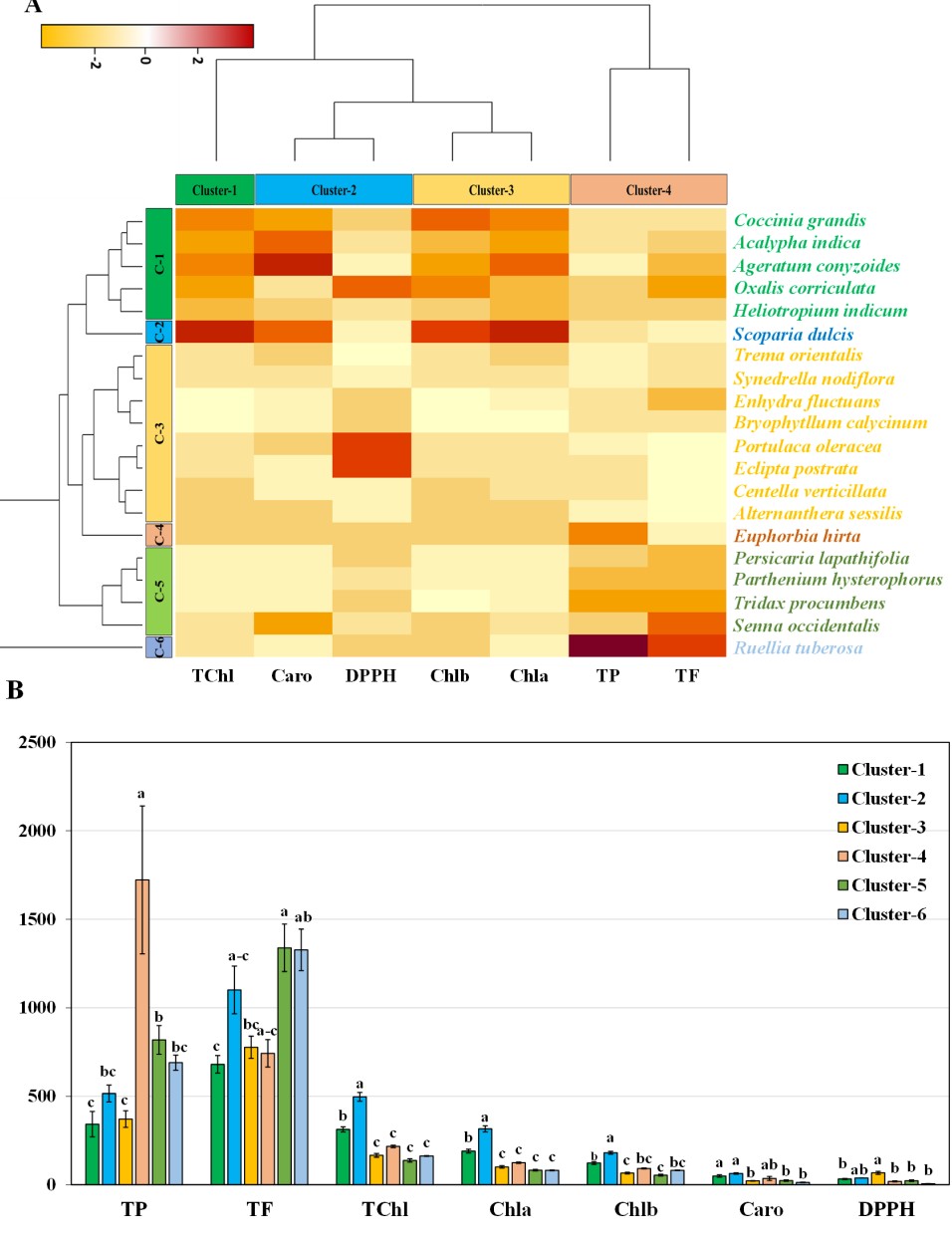

**Figure 5 Hierarchical clustering and bar charts were generated to display the relationships between species and traits.** (A) Heatmap presenting scaled average values for seven stem traits, with colours indicating the relative scale ($-3$ to $+3$) from the data standardization of pigments, antioxidants, and $IC_{50}$ values. The species and traits were grouped into six (rows) and four (columns) clusters, respectively. (B) Bar graph showing comparative cluster analysis of the stems of selected species. The studied traits were Chla (chlorophyll $a$), Chlb (chlorophyll $b$), TChl (total chlorophyll), Caro (carotenoids), PC (phenolic), FC (flavonoid), and DPPH ($IC_{50}$ value for scavenging DPPH free radical). The data are presented as the means of 3 replicates $\pm$SEs, with a sample size of $n = 3$. Statistical analysis was performed using one-way ANOVA with Tukey's post hoc test; different letters (a, b, c, ........., *etc.*) indicate statistically significant differences ($p < 0.05$).

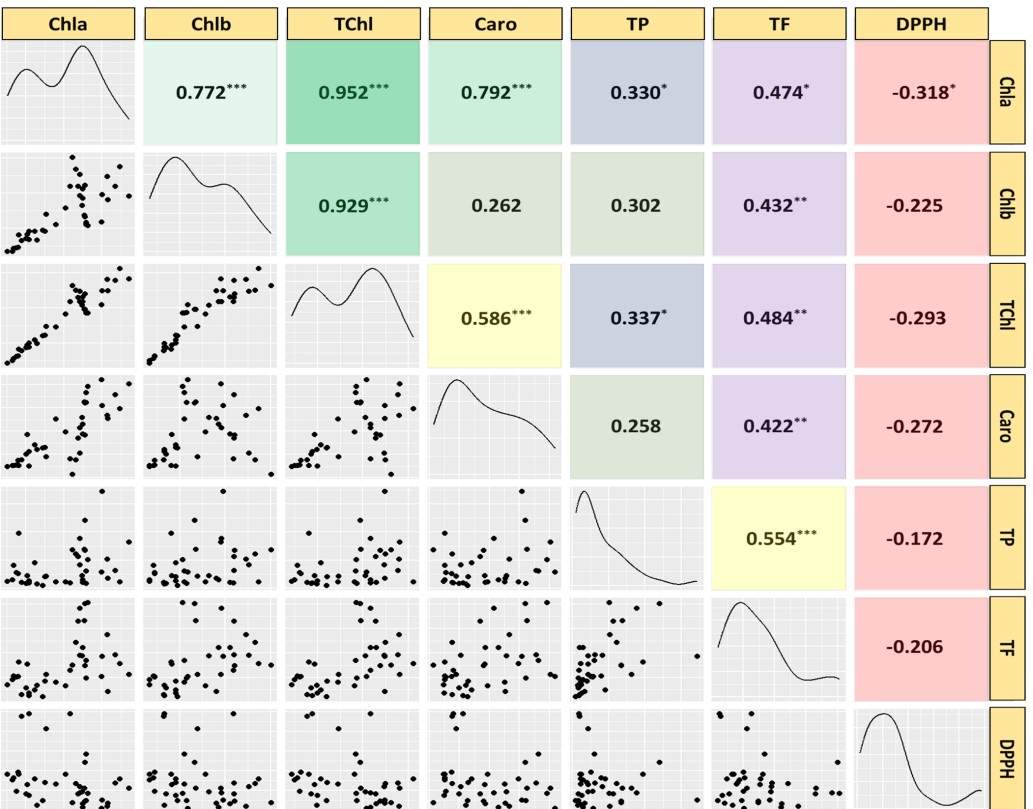

**Figure 6  Scatterplot, correlation matrix and distribution of chlorophyll, carotenoids, phenolics, flavonoids, IC$_{50}$ value to scavenge free radicals of the 20 studied medicinal weed species.** Chla, chlorophyll *a*; Chlb, chlorophyll *b*; TChl, total chlorophyll; Caro, carotenoids; PC, phenolics; FC, flavonoids; DPPH, IC$_{50}$ value to scavenge free radicals) of the 20 studied medicinal weed species. The red and green boxes in the upper panel denote positive and negative correlations, respectively. A higher coefficient is associated with an increase in color intensity. The distribution histogram of related species is displayed in the diagonal panel. Correlations were found among most pigments and phytochemical traits, except for Caro and Chlb, TP and Chlb, and TP and Caro. While there were nonsignificant negative correlations with the other parameters, the radical scavenging potential (DPPH) showed significant negative correlations with the Chla concentration. Taken together, these findings imply that the contents of both of the investigated phytochemicals increase as the pigment content increases, and the opposite occurs. However, the contents of every pigment and antioxidant decrease as the DPPH radical concentration increases.

## Principal component analysis

This principal component analysis (PCA) study involved 20 species and seven traits measured in leaf and stem parts. The analysis revealed that seven principal components with eigenvalues greater than one collectively explained 70.07% of the total variability. The biplot representation of the PC1 and PC2 scores demonstrated that PC1 alone accounted for 14.82% of the variance, while PC2 contributed a substantial 55.25%. The loadings associated with each principal component provided insights into the relationships between the original traits and the newly derived components (Fig. 7).

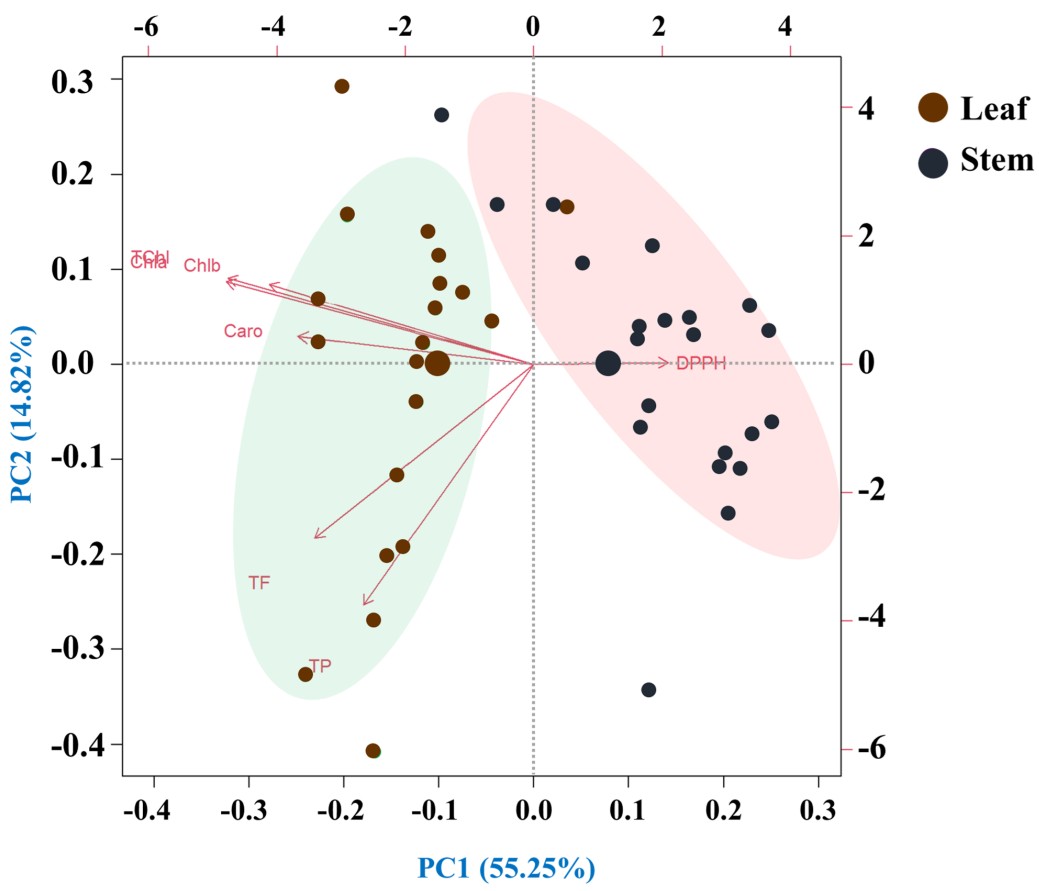

**Figure 7** **Principal component analysis (PCA) of pigments, antioxidant properties and free radical scavenging potential (arrows) of leaves and rhizomes of different species (points).** Species are distributed along distinct ordinates according to their differences from one another. The strength of each vector's (line's) contribution to each PC is indicated by its magnitude. Positive or negative interactions between the studied species are displayed by the angles between the vectors derived from the middle point of the biplots. There is an extremely strong positive correlation between the close variables (vectors) that form small angles. The PCA scree plot, located at the bottom, displays the variance proportion of each principal component. Traits: (Chla, chlorophyll *a*; Chlb, chlorophyll *b*; TChl, total chlorophyll; Caro, carotenoids; PC, phenolics; FC, flavonoids; DPPH, IC$_{50}$ value for scavenging free radicals).

## DISCUSSION

The increasing acceptance of plant-based medications for primary care is a result of their cost effectiveness (*Fabricant & Farnsworth, 2001*). The ability of plant parts to scavenge free radicals relies on a diverse array of bioactive compounds, including phenolics, flavonoids, vitamin C, and glutathione (*Dhalaria et al., 2020*). Chlorophyll and carotenoids in plants have health benefits like fighting cancer and acting as antioxidants. This motivates research to find plants rich in these compounds for herbal medicine (*Ghosh et al., 2018*). Chlorophylls demonstrate a diverse array of advantageous properties, such as antimutagenic, antigenotoxic, and anti-obesity effects (*Martins et al., 2023*). The distinctive chemical composition of chlorophyll enables it to combat detrimental free

radicals, reduce DNA damage, and regulate cellular mechanisms implicated in disease progression. Additionally, its hydrophobic side chains facilitate engagement with biological membranes, impacting cellular absorption and signaling pathways (*Zepka, Jacob-Lopes & Roca, 2019*; *Perez-Galvez, Viera & Roca, 2017*). Carotenoids, the well-known lipophilic isoprenoid compounds, produce biologically active molecules in both plants (hormones, retrograde signals) and animals (retinoids) after enzymatic breakage (*Rodrıguez-Concepción et al., 2018*).

Phytochemical or natural antioxidants are secondary compounds found in plants. They are particular substances that safeguard cells in humans, animals, and plants from the harmful impacts of free radicals, also known as reactive oxygen species (ROS) (*Dai & Mumper, 2010*). Examples of antioxidants include phenolic acids, flavonoids, and carotenoids, which are produced by plants to support their survival (*Apak et al., 2007*). One of the key attributes of polyphenols is their ability to scavenge radicals, contributing to antioxidant properties, and their capacity to engage with proteins (*Ozcan et al., 2014*). These properties are crucial as they enable the compounds to effectively absorb and neutralize free radicals, extinguish both singlet and triplet oxygen, and break down peroxide molecules (*Hasan et al., 2008*). Plants that contain phenolic compounds with aromatic rings have several beneficial characteristics, including the ability to chelate metals and act as antioxidants (*Bhatt & Negi, 2012*; *Khoddami, Wilkes & Roberts, 2013*). To transform from a free radical into a stable diamagnetic molecule, DPPH can absorb an electron or hydrogen radical. This free radical is scavenged by the antioxidants in the sample. This method is widely used to evaluate the DPPH and free radical scavenging ability of natural antioxidants (*Islam et al., 2018*; *Maharana et al., 2010*). The DPPH method is simple, fast, and convenient, making it ideal for screening a large number of samples for radical scavenging ability. It remains unaffected by sample polarity (*Marxen et al., 2007*).

Based on the study result, *O. corniculata* plants had an excellent chlorophyll *b* content, and *S. dulcis* species had the highest chlorophyll *a* and total chlorophyll contents. The relevant literature *Begum & Bora (2018)* confirms high total chlorophyll levels in *A. sessilis*, *Centella asiatica* L., and *O. corniculata* plants. *A. indica* plants possess best carotenoid content among the studied species. However, according to another study, the total carotenoid content found in *A. indica* is reported as 0.513 mg/g tissue, which shows significant variation from the results obtained in the study (*Ghosh et al., 2018*). Variations in plant result may arise from either climatic and soil conditions or the age of the plant. Researchers have also found significant amounts of carotenoids in *O. corniculata* leaves (*Zeb & Imran, 2019*). Another report revealed significant levels of carotenoids and chlorophyll in *A. indica*, *T. procumbens*, and *E. hirta* (*Ghosh et al., 2018*).

Among twenty medicinal weeds, significant variations in phenolic content were noted. *E. hirta* displayed the highest levels in both leaf and stem parts, while *P. oleracea* and *O. corniculata* showed the lowest content. *Tran et al. (2020)* suggested the presence of potent phenolics in *E. hirta* which aligns with our results. Another study revealed a greater phenolic content in *R. tuberosa* (*Thi Pham et al., 2022*). Moreover, the flavonoid content-one of the most significant antioxidants-was evident in every species studied. *P. lapathifolia*, *S. occidentalis*, *R. tuberosa* and *T. procumbens* were the most abundant in flavonoid content.

These findings are further supported by previous research (*Yakubu et al., 2021*; *Thi Pham et al., 2022*).

*Thi Pham et al. (2022)* suggested a considerably lower $IC_{50}$ in the *R. tuberosa* plant, which has the highest DPPH scavenging potential. Among the twenty species *R. tuberosa* exhibited the highest phenolic content, and *H. indicum* displayed the highest chlorophyll and carotenoid contents, which are related to its radical scavenging capacity. Furthermore, although *P. hysterophorus* possesses an average amount of pigments and antioxidants, having a lower $IC_{50}$ value indicates a greater free radical scavenging capacity. This is because additional substances that were not examined in the plant extracts may enhance the antioxidant potential through interactions (*Surendraraj, Farvin & Anandan, 2013*). Correlation studies unveil connections between parameters. They assist breeders in selecting plant varieties with desired traits (*Ghafoor et al., 2013*; *Mohi-Ud-Din et al., 2021*). After careful examination, the correlation of the genotypes showed some fascinating patterns. Almost all the traits examined in this study exhibited strong correlations, suggesting that modifications to one trait would affect others. The rhcoclust R package was used for clustering traits and genotypes, known for its strong outlier handling. These matrices facilitate targeted selection based on specific traits (*Hasan, Badsha & Mollah, 2020*).

The high phenol, flavonoid, and bioactive pigment levels in the twenty plants contributed to their robust free radical scavenging activity, positioning them as potential natural antioxidants. However, the leaves performed better than the stems for every attribute. The reason can be that leaves are specialized for capturing sunlight and turning it into energy through photosynthesis (*Johnson, 2016*). This crucial function demands a diverse range of biochemicals like chlorophyll, carotenoids, and various phenolic compounds. Moreover, leaves act as key sites for gas exchange and water regulation, requiring the presence of biochemicals involved in these vital processes. These include compounds that regulate stomatal function and antioxidants that safeguard against oxidative stress induced by intense light exposure (*Roth-Nebelsick & Krause, 2023*).

The highest performing species, *H. indicum,* demonstrated superior antioxidant potential, which was supported by the findings of *Wani, Tolu & Wahid (2018)*. *Ghosh et al. (2020)* retrieved that 70% ethanolic extract of *E. hirta* possesses the highest number of bioactive compounds among the five selected medicinal weeds. *E. hirta* showed the most favorable performance across all studied traits within row cluster-3. Again, leaf clusters 1 and 2 showed no significant differences in pigment content, as indicated by the high flavonoid content in cluster 1 and the high phenolic content in cluster 2. Among the stem samples, cluster 2 had the highest pigment content, while clusters 4, 5, and 6 exhibited superior free radical scavenging potentials with elevated phenolic and flavonoid contents. In PCA, angles in biplots demonstrate trait correlations; acute angles signify positive correlations, obtuse angles indicate negative correlations, and a 90° angle implies independence (*Abdi & Williams, 2010*; *Bahrami, Arzani & Karimi, 2014*; *Mohi-Ud-Din et al., 2021*). Nonetheless, our findings clearly showed that a trait pair's correlations were well coordinated with the same trait pair's contribution to the PCA biplot and the estimated values of the vector angles.

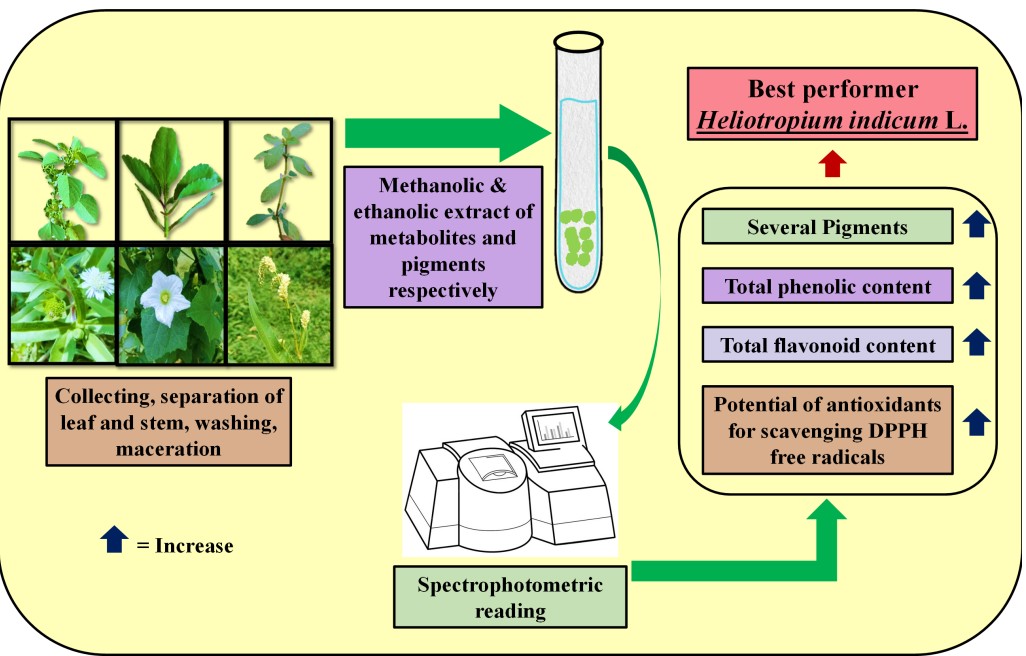

**Figure 8 Simplified flow chart for determination of several pigments, phytochemical properties and free radical scavenging potential in medicinal weeds.** Photo credit (plants and plant parts): Mousumi Jahan Sumi (First author); Image credit: the UV spectrophotometer archive at Freepik (PearPran, Premium license, https://www.freepik.com/premium-vector/uv-vis-spectrophotometer-diagram-experiment-setup-lab-outline-vector-illustration_29972735.htm#fromView=search&page=1&position=12&uuid=275cc435-5.4e-44e8-a923-2bc070768be5).

Finally, the study uncovers phytochemicals, antioxidants, and pigments in tested weed species but lacks in-depth exploration of their molecular antioxidant mechanisms or synergistic interactions. Further research is needed to optimize weed-based treatments for primary care.

## CONCLUSIONS

The study examined plant-part characteristics in species of medicinal weeds and showed that leaves have consistently greater concentrations of flavonoids, phenolic acids, carotenoid compounds, and chlorophyll than stems. As demonstrated by the $IC_{50}$ values, the leaves had a superior capacity to scavenge antioxidants, which highlights their importance in the fight against oxidative stress. Of the twenty species that were studied, *H. indicum* was the best individual (Fig. 8). Principal component analysis (PCA) and hierarchical clustering provide information about genotype similarities. These weeds could be eco-friendly alternatives to synthetic pesticides. By studying weeds' antioxidants in Bangladesh's south, we discovered new medicinal resources for potential nutraceutical industries. Ultimately, our endeavor aims to facilitate the utilization of natural medicines derived from weeds, providing cost-effective and sustainable solutions for the healthcare and pharmaceutical sectors. Future research could focus on elucidating the specific mechanisms underlying the

therapeutic properties of medicinal weed species and exploring their potential applications in targeted healthcare interventions.

## ACKNOWLEDGEMENTS

The authors extend their appreciation to Khulna University, Khulna, Bangladesh for providing the laboratory facility.

### Funding

This research was supported by Khulna Agricultural University, Khulna 9100, Bangladesh. The study was also supported by the Operational Program Integrated Infrastructure within the project: Demand-driven Research for the Sustainable and Innovative Food, Drive4SIFood 313011V336, co-financed by the European Regional Development Fund. Taif University, Saudi Arabia, Project No. (TU-DSPP-2024-07) also funded this research and they provided necessary laboratory facilities. The Ministry of Education, Youth and Sports of the Czech Republic (S grant of MSMT CR) provided financial support to conduct the study. The funders had no role in study design, data collection and analysis, decision to publish, or preparation of the manuscript.

### Grant Disclosures

The following grant information was disclosed by the authors:
Khulna Agricultural University, Khulna 9100, Bangladesh.
The Operational Program Integrated Infrastructure within the project: Demand-driven Research for the Sustainable and Innovative Food, Drive4SIFood 313011V336.
The European Regional Development Fund.
Taif University, Saudi Arabia: TU-DSPP-2024-07.
The Ministry of Education, Youth and Sports of the Czech Republic (S grant of MSMT CR).

### Competing Interests

The authors declare that there is no conflict of interest in the article.

### Author Contributions

- Mousumi Jahan Sumi conceived and designed the experiments, performed the experiments, analyzed the data, prepared figures and/or tables, authored or reviewed drafts of the article, and approved the final draft.
- Samia Binta Zaman performed the experiments, prepared figures and/or tables, authored of the article, and approved the final draft.
- Shahin Imran conceived and designed the experiments, supervision, analyzed the data, prepared figures and/or tables, authored or reviewed drafts of the article, and approved the final draft.
- Prosenjit Sarker performed the experiments, prepared figures and/or tables, and approved the final draft.
- Mohammad Saidur Rhaman authored or reviewed drafts of the article and approved the final draft.
- Ahmed Gaber analyzed and curated data, prepared figures and/or tables, funding acquisition, authored or reviewed drafts of the article, and approved the final draft.
- Milan Skalicky supervision and project administration analyzed and curated data, prepared figures and/or tables, funding acquisition, authored or reviewed drafts of the article, and approved the final draft.
- Debojyoti Moulick analyzed and curated data, prepared figures and/or tables, funding acquisition, authored or reviewed drafts of the article, and approved the final draft.
- Akbar Hossain supervision and project administration, analyzed and curation of data, analyzed the data, prepared figures and/or tables, funding acquisition, authored or reviewed drafts of the article, and approved the final draft.

## Data Availability

The raw measurements are available in the Supplementary Files.

## Supplemental Information

Supplemental information for this article can be found online at http://dx.doi.org/10.7717/peerj.17698#supplemental-information.

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
