# Peer review of "An investigation of the pigments, antioxidants and free radical scavenging potential of twenty medicinal weeds found in the southern part of Bangladesh"

_PeerJ, doi:10.7717/peerj.17698_

## Round 0.1 · original submission · Major Revisions

Dear Authors,

Please revise your article as advised by the Reviewers.

Reviewers 2 & 3 have requested that you cite specific references. You may add them if you believe they are especially relevant. However, I do not expect you to include these citations, and if you do not include them, this will not influence my decision.

Regards

·

Basic reporting

Title: An investigation of the pigments, antioxidants and free radical scavenging potential of medicinal weeds
Thank you very much for sharing your article to review. My comments are below-
A. Abstract
It should be rewritten to focus on background, methods, results, and conclusions.
B. Introduction
Discuss your research questions and why your work is important. What is lacking will fulfill your research.

C. Do you have HPLC data for analyzing phytochemicals from leaves and steam extract? For the exact information, this data is very important.

Line 125-Materials & Methods
Sampling and data collection
1. How do you extract from plant leaves and steam? Did you sundry? Did you make powder? Which solvent did you use to extract? Explain precisely.
2. Clearly write the data availability section

Figure 1
Increase the font size for better visibility
Figure 2
1. Increase the front of the leaf and steam indication box
2. Write the meaning of small a, b, c… etc.
3. Indicate the statistics on the graph appropriately
Figure 3
Same comments as Figure 2
Figure 4
Make the figure B-H are unique. Add statistics to each graph
Figure 5
Make the figure B-H are unique. Add statistics to each graph.

Experimental design

Should be improved

Validity of the findings

Should be improved

·

Basic reporting

> The article includes sufficient introduction and background to demonstrate how the work fits into the broader field of knowledge. Relevant prior literature should be appropriately referenced.
However, I found another article published by Ghosh et al. 2020 (Ghosh P, Das C, Biswas S et al. Phytochemical composition analysis and evaluation of in vitro medicinal properties and cytotoxicity of five wild weeds: A comparative study [version 1; peer review: 1 approved, 2 approved with reservations]. F1000Research 2020, 9:493 (https://doi.org/10.12688/f1000research.22966.1) is relevant to the study since they share some common weeds and a set of phytochemical data and hence it can be cited in the proper place of introduction and discussion.
Similarly, the authors are advised to check, for any other similar study/studies and if found, they need to be discussed in line with the study.

Experimental design

> The research gap needs to be properly discussed concerning the previous study.
Otherwise, it is found to be okay.

Validity of the findings

Whether the sampling was made from the same plants from the University campus. or different plants?
Since the work is confined to some important parameters like phenolics, flavonoids and pigments, DPPH assays only, they are advised to check the variability within same sample, among several samples if collected.
In supplementary data, the duly signed pictures of the herbariums of the different weeds need to be provided.

Additional comments

The research paper is entitled “An investigation of the pigments, antioxidants and free radical scavenging potential of medicinal weeds” under review offers a comprehensive examination of the application of biochemical and in-depth statistical techniques. This is a very extensive work and presented in an appropriate way in lucid languages that would certainly attract a scientific audience worldwide. Here the authors have made biochemical analysis of total polyphenolics, flavonoids, pigments and antioxidant power of different medicinal weeds and tried to explore the data in all possible statistical techniques. Use of box plot, PCA analysis etc are truly praiseworthy. However, some points I need to mention, given below:

According to me, the title should be changed to “An investigation of the pigments, antioxidants and free radical scavenging potential of twenty medicinal weeds found in the southern part of Bangladesh” for better understanding.

However, during close inspection, I found some data was presented in an abstract and also in some positions in text containing different numbers of digits after point (line no 41 to 45). That needs correction.

Whether the spectrophotometric data are taken using the same spectrophotometer as mentioned in the pigment analysis section or not? If different then please mention their details. Otherwise, in the material section, instruments could be introduced where all the instruments used in the study may be documented.

In the sampling section, the author mentioned they collected 4 plants of a single species from the campus. Whether the 4 plant samples mixed to get the data?
The study had been performed with fresh materials, as evident from FW data. Why were the studies not done in a dry sample?
Antioxidant power has been checked only by DPPH assay. It is only a free radical scavenging assay. FRAP assay could be performed in addition to the DPPH assay to understand the antioxidant power.

Reviewer 3 ·

Basic reporting

I have reviewed the MS “An investigation of the pigments, antioxidants and free radical scavenging potential of medicinal weeds”
The MS is well presented but here I offer some improvements
Follow the rules of taxonomy in documenting scientific names in the MS.
Some recent findings need to be included in MS like;

https://doi.org/10.1016/j.bcab.2024.103058
https://doi.org/10.1016/j.arabjc.2023.104653
https://www.pakbs.org/pjbot/papers/1641325052.pdf
and
https://openurl.ebsco.com/EPDB%3Agcd%3A12%3A9820748/detailv2?sid=ebsco%3Aplink%3Ascholar&id=ebsco%3Agcd%3A156264838&crl=f
and old references may be removed
Rationale not provided
There is room for clear and unambiguous, professional English in MS.
All scientific names in MS as well as in raw data must be correct e.g., Oxalis corniculate spelled wrongly in raw data.
Figures need to be inserted with good resolution, Complete raw data not shared. Raw values for each replicate need to be provided. e.g., Spectrophotometer readings.

Experimental design

The protocol was slightly modified after Lichtenthaler (Lichtenthaler, 1987). What modifications? State briefly
All equations need to be written using equation editor
Discussion need to be revised in a scientific way; 1st describe your own results then support with literature. Discussion needs to be elaborated
Crossmatch all references
Figure high resolution without blur. Letters should be clearly visible
Statistics need to be rechecked. Some error bars not visible?
p<0.05 p should be italics in all MS
In figures scientific abbreviations may not be used
The MS falls within aims and scope of the journal.
Research question not well defined,
Rigorous investigation performed by the authors
The authors should briefly discuss the modifications in the methods

Validity of the findings

Impact and novelty not stated well.
Recent literature needs to be cited.
In conclusion, authors need to add the prominent weeds and their impact/usage.
Statistics needs to be rechecked. Raw data need attention. Raw values for each replicate need to be provided. e.g., Spectrophotometer readings.

·

Basic reporting

Manuscript is not clear, lack of professional English used throughout

Experimental design

Research question required clarification and have knowledge gap

Validity of the findings

All underlying data needs improvements

Additional comments

Although the manuscript is well-prepared, there are some major problems that needs to be resolved.
The manuscript's grammar and language should be improved. There is a certain repetition in the abstract.
The last section of the introduction needs to include the rationale for the research.
Results pertaining to phytochemicals should be used in the methodological sections. Improved figures are needed in the results area for statistical analysis and better resolution.
Authors are strongly advised to consider currently published manuscripts to narrate the findings of a study.
The discussion is poorly written.
It is recommended that authors include some graphic proof that supports up their conclusions.

·

Basic reporting

The manuscript reported the measurement of chlorophyll, carotenoid, polyphenol, flavonoid and antioxidants in 20 weed plants which might be used for medicinal usage. The results were systematically compared and statistically analyzed in depth. Although the experiment methods were not state-of-art, even out of date, the results are interesting and provide useful information for the usage of these medical weeds.
About the reference, I noticed that authors cited many medical usage of these weed plants in the introduction section. This is good, however, I think the authors could cite more references reporting the metabolite analysis of these plants in the discussion section and compare their data with results of this study. Furthermore, the authors should also discuss the possible reason why some plants have more polyphenol and flavonoid contents. This discussion will make the research more interesting.

Experimental design

The experiments are well performed. I have no objections.

Validity of the findings

The results were well analzyed, I have no objections.

Additional comments

The papar was well wroten. However, some places still have to be improved. I listed some errors as following, but the authors have to carefully check whether other places are correct or not.
1. title: "medical weeds" should be more specific, "twenty medical weeds" is better.
2. Line 69: "However" should be changed to "Indeed"
3. Line 284 and 422: "Rhizomes" shoud be changed to "Stems"
4. Line 487-488: "Also funded by" should be deleted

Reviewer 6 ·

Basic reporting

The manuscript is well-structured, presenting a clear objective to explore the phytochemical composition and antioxidant potential of medicinal weeds. The literature review is extensive, linking the importance of weeds not only as nuisances but also as potential sources of natural antioxidants. The methods for determining pigments, phenolic, and flavonoid contents, as well as the antioxidant capacity using DPPH assay, are clearly described, aligning with standard practices in the field. The use of Mini Tab 17.3 for statistical analysis and various R packages for data visualization and analysis indicates a thorough approach to data handling and representation. The introduction and background offer a comprehensive overview of the importance of weeds as medicinal resources, setting a strong foundation for the study's objectives. However, the manuscript would benefit from including more recent studies to contextualize its findings within the latest research advancements.

Experimental design

The experimental design is methodologically sound, with a clear definition of the research objective and a detailed description of the methods used for sampling, data collection, and analysis. The choice of twenty different species of medicinal weeds and the inclusion of both leaf and stem parts for analysis reflect a comprehensive approach to assessing the phytochemical diversity within these plants.

The study design, including the sampling strategy and analytical methods, follows a rigorous scientific approach, allowing for a detailed exploration of phytochemical and antioxidant properties. While the use of traditional methods for pigment and antioxidant content analysis is valid, the manuscript could have discussed the rationale behind the selection of these specific species and parts more thoroughly. Additionally, incorporating more advanced complementary analytical techniques, such as liquid chromatography-mass spectrometry (LC-MS) for phytochemical profiling, could enhance the depth of phytochemical analysis.

Validity of the findings

The findings present a comprehensive overview of the pigment, phenolic, flavonoid contents, and antioxidant capacities of the studied weeds, contributing valuable insights into their potential medicinal properties. The statistical analysis appears robust, with significant differences highlighted among the species and plant parts. The use of hierarchical clustering and PCA further aids in understanding the data's complexity and the relationships between different phytochemical traits.

The thorough statistical analysis and the use of advanced data visualization techniques strengthen the validity of the findings, providing a clear picture of the potential of these weeds as sources of natural antioxidants. While the findings are compelling, the manuscript could benefit from a more detailed discussion of how these results compare with those reported by existing literature. For instance, discussing how the antioxidant capacities of these weeds compare with other known medicinal plants could offer a more nuanced understanding of their potential efficacy.

Additional comments

The manuscript makes a notable contribution to the field of natural product research, particularly in the exploration of weeds as a source of antioxidants. The detailed analytical and statistical methods used in the study provide a solid foundation for the findings. However, incorporating more recent references and possibly expanding the analytical methods to include more advanced techniques could enhance the study's impact.
I recommend the publication of this manuscript with minor revisions. Incorporating the latest studies and possibly discussing some additional items (see below) on study findings and limitations would further strengthen the manuscript.

More Discussion on Findings:
1. Comparative Analysis: I encourage the authors to deepen their analysis by comparing their findings with those of similar studies. For example, they could discuss how the antioxidant activities of the weeds studied compare to those of well-known medicinal plants, considering both traditional uses and scientifically established data. This would provide a richer context for the significance of their findings.
2. Phytochemical Diversity: The authors may discuss the implications of the phytochemical diversity observed across different species and plant parts for pharmaceutical and nutraceutical applications. This discussion could include potential synergistic effects of the phytochemicals identified and their role in the antioxidant capacity of the plants.
3. Species-Specific Insights: The manuscript could benefit from a more detailed discussion on species that exhibited particularly high levels of antioxidants or unique phytochemical profiles. Exploring the potential reasons for these findings, such as genetic factors, environmental conditions, or evolutionary adaptations, would add depth to the analysis.
4. Clinical Relevance: The authors may discuss the clinical relevance of their findings, considering the bioavailability, metabolism, and potential health benefits of the identified phytochemicals. This could involve referencing studies that have explored the efficacy of these compounds in vivo or in clinical settings.

Addressing Limitations:
1. Methodological Limitations: While the methods used are sound, the authors should discuss any limitations associated with their experimental design. For example, the extraction methods used might not capture all bioactive compounds, or the assays for antioxidant activity might not fully represent in vivo efficacy. The impact of these limitations on the findings should be acknowledged.
2. Species Selection: The rationale behind the selection of the twenty species should be discussed, including any biases this might introduce into the study. For example, if the selection was based on traditional uses, the authors should discuss how this might influence the generalizability of their findings to other medicinal weeds.
3. Reproducibility and Scalability: Discuss the challenges associated with reproducing the study's findings on a larger scale, including variability in phytochemical content due to environmental factors and the potential impact on the antioxidant activities observed.
4. Future Research Directions: The authors should outline specific areas where further research is needed, such as investigating the mechanisms of action of the identified phytochemicals, exploring their efficacy in disease models, or assessing the impact of cultivation practices on phytochemical profiles.

By expanding the discussion in these areas and explicitly addressing the study's limitations, the manuscript would provide a more comprehensive understanding of the significance of the findings and their implications for the use of medicinal weeds in healthcare. This approach would also highlight the potential for future research to build on the foundation laid by this study, further advancing our understanding of the role of weeds in natural product research.

---

## Round 0.2 · Minor Revisions

Dear Authors,

Please improve your manuscript as advised by the reviewers. Please remember that language of the manuscript still needs improvement.

·

Basic reporting

Title: An investigation of the pigments, antioxidants and free radical scavenging potential of twenty medicinal weeds found in the southern part of Bangladesh
Thank you for sharing this article for review. This article will be considered after some revision. My comments are below-
The writing quality is too poor. Writing quality must be improved.
Materials & Methods
Sampling and data collection
1. After collecting samples weeds, who confirmed your sampling is correct? Did you confirm this with the help of a plant biologist? If so, please add this information.
2. How did you extract? Add this information.
3. Add information about the solvents that you used.
Line 152, write as Two hundred microliters (200 μL)
Line 168-169, improve the quality of your writing as below-
The 3g fresh leaf and stem samples were homogenized in 30 mL of 99.9% methanol using a mortar and pestle.
Line 170, add comma “,” before and, also scavenging spelling is incorrect, delete of before centrifugation. The sentence should be like that-
The supernatant 170 extracts were used for phenol, flavonoid, and DPPH radical scavenging capacity quantification 171 after 30 minutes of darkness and 5 minutes centrifugation at 15000 rpm.
Line 176, write as of 176
Line 183, add space between 2mLNaOH
Line 187, this sentence should be like that
total flavonoid content (TFC) was measured as µg of equivalent catechin per gram of fresh extract
Lines 190-191, do not make sense, rewrite
Line 192, The violet-colored free radical is called DPPH, does not make sense; define DPPH appropriately.
Line 217, use the article “a” before highly.
In Fig. 4(A) and 5 (B), a or b overlaps. This overlap should be minimized.

Experimental design

Must be improved. Writing quality is poor

Validity of the findings

Should be improved. Writing quality is poor

Reviewer 3 ·

Basic reporting

Although the authors have tried to improve the MS but still there is gap to cover.
There are still issues in English. Expression of English needs to be addressed. Avoid we analyzed… or In our study,… and We also found… and more…

Scientific names should be given with author citation only at 1st mention in the document then genus name be abbreviated and no author citation. Consult some taxonomist please.

Formatting issues at many places, Phytomedicine :

Experimental design

Quality of all pictures still poor/blur

Validity of the findings

The authors have improved

Reviewer 6 ·

Basic reporting

No more comments.

Experimental design

No more comments.

Validity of the findings

No more comments.

Additional comments

The authors have satisfactorily addressed all of my questions. I recommend the manuscript for publication.

Reviewer 7 ·

Basic reporting

Your manuscript entitled 'Exploring the Pigments, Antioxidants, and Free Radical Scavenging Potential of Twenty Medicinal Weeds from Southern Bangladesh' is well-written and offers valuable insights into the biochemical properties of indigenous weed species. We suggest further improvements for your consideration
1. How were the criteria established for the selection and processing of the twenty different weed species? Additionally, could the methodology section be expanded to provide more detail on the process of sample collection, handling, and preparation to ensure transparency and reproducibility?
2. According to Line 151 and 152 The protocol was slightly modified after Lichtenthaler (Lichtenthaler, 1987). justify?
3. Could the results be presented in the form of percentages for each section to facilitate clearer interpretation and comparison?
4. Should the authors specify any transcription levels for antioxidants in their study?
5. Could the discussion section be enhanced by incorporating a mechanistic approach to elucidate the underlying processes driving the observed antioxidant activity of the studied medicinal weed species?
6. Could you please clarify the referencing style specified in the journal's instructions so that the citation format can be aligned accordingly?
7. Please contact a fluent English speaker who can help refine the language in your document for improved clarity and fluency.

Experimental design

Could you elaborate on the rationale behind the biochemical processing methods and sampling preservation techniques outlined in the methodology section for clarity and reproducibility?

Validity of the findings

Your manuscript entitled 'Exploring the Pigments, Antioxidants, and Free Radical Scavenging Potential of Twenty Medicinal Weeds from Southern Bangladesh' is well-written and offers valuable insights into the biochemical properties of indigenous weed species. We suggest further improvements for your consideration for further perusal

Additional comments

Recommended for publication after amendments

---

## Round 0.3 · accepted · Accept

Authors have improved their manuscript. Therefore, this manuscript may be accepted for publication.